# Hard- and Software Controlled Complex for Gas-Strain Monitoring of Transition Zones

**DOI:** 10.3390/s24082602

**Published:** 2024-04-18

**Authors:** Grigory Dolgikh, Mariia Bovsun, Stanislav Dolgikh, Igor Stepochkin, Vladimir Chupin, Andrey Yatsuk

**Affiliations:** V.I. Il’ichev Pacific Oceanological Institute, Far Eastern Branch Russian Academy of Sciences, 690041 Vladivostok, Russia; dolgikh@poi.dvo.ru (G.D.); bovsun.ma@poi.dvo.ru (M.B.); sdolgikh@poi.dvo.ru (S.D.); stepochkin.ie@poi.dvo.ru (I.S.); yatsuk@poi.dvo.ru (A.Y.)

**Keywords:** gas-strain monitoring, laser strainmeter, gas analyzer, greenhouse gases, transition zones

## Abstract

The article describes a hard- and software controlled complex for gas-strain monitoring, consisting of stationary laser strainmeters and a laser nanobarograph, a stationary gas analyzer, and a weather station installed at Shultz Cape in the Sea of Japan; and a mobile shipboard complex, consisting of a gas analyzer and a weather station installed in a scientific research vessel. In the course of trial methodological measurements on these systems, general patterns were identified in the dynamics of greenhouse gases and deformation of the Earth’s crust in the range of diurnal and semi-diurnal tides, and also in the range of ultra-low frequencies, caused by atmospheric wave processes and, possibly, individual tones of the Earth’s eigen oscillations.

## 1. Introduction

Recently, with increasing alarm, we have been following the global climate changes taking place on Earth. These global climate changes are especially vivid in the observed meteorological parameters, such as greenhouse gases, surface air temperature, precipitation, general atmospheric circulation, the state of the land and world ocean cryosphere, and extreme climate conditions.

Atmospheric concentrations of three greenhouse gases, from both natural and anthropogenic sources, have increased significantly since the pre-industrial era: CO_2_—by 46%, CH_4_—by 157%, and N_2_O—by 22%.

According to the World Meteorological Organization, 2015–2020 were the warmest six years, and 2011–2020 was the warmest decade in the history of observations. Since the 1980s, each subsequent decade has been warmer than any previous decade after 1850. The current average global surface air temperature is approximately 14.9 °C, which is 1.2 °C higher than in the pre-industrial era. Average warming rates of surface air during 1976–2020 amounted to 0.18 °C/10 years on the global scale, and during this period alone, the global temperature increased by 0.8 °C. Temperature has been increasing especially rapidly in the North Polar region, where for over 30 years (1991–2020) the linear increase in average annual temperature was about 2.64 °C.

In recent decades, the amount of global precipitation over land has been increasing, and the rate of this increase is accelerating. This increase, according to various data sources, is estimated at 5–10 mm/year per decade against the background of significant variability year to year and regional heterogeneity. In the Northern Hemisphere, the total number of extratropical cyclones has grown, with a decrease in the number of intense cyclones; and in the Southern Hemisphere, the number of intense extratropical cyclones has increased. It is also probable that global weakening of surface winds over land has occurred, which is particularly noticeable in the Northern Hemisphere.

The temperature of the ocean surface continues to increase, extending to its deeper layers. In 2021, the significance of this trend has decreased to 12.7% over a decade. The rise in the level of the World Ocean is accelerating. Over the entire twentieth century it amounted to approximately 17 cm, and then increased from 3.04 mm/year in the period of 1997–2006 to 4.36 mm/year in 2007–2016. Since the early 1990s, the average sea level has risen by about 90 mm. In the Arctic Ocean, the sea ice area continues to decrease, most intensely in September, amounting to 13.1% per decade in 2020 as compared to the 1981–2010 average. Arctic sea ice is becoming younger and thinner on average. In the surface layer of the ocean, the pH decreases, mainly as a result of carbon dioxide absorption. According to estimates made based on observation series for 15 years or more, the rate of pH decline (i.e., acidification of the waters of the World Ocean) was 0.017–0.027 units per decade.

The trend of recent decades towards an increase in the number of dangerous hydrometeorological phenomena continues. Moreover, according to statistics from insurance agencies that assess the degree of danger based on the amount of caused material damage, the main contribution to this increase is made by floods (precipitation) and weather disasters caused primarily by wind (hurricanes, storms, whirlwinds, tornadoes, etc.). In recent years, heat waves have become the reason for the largest number of deaths caused by severe weather events; the maximum economic losses were associated with tropical cyclones.

The sixth session of the United Nations noted that in order to keep the increase in global temperature at a level of no more than 1.5 °C, like at the dawn of the industrial era, it is necessary to take the following measures. By no later than 2025, the level of global greenhouse gas emissions must begin to reduce; levels must reduce by about 43 percent by 2030; by 60 percent by 2035, relative to the level of 2019; and net zero emissions must be achieved by no later than 2050.

With temperatures rising and catastrophic processes increasing, the volume of natural emissions is also increasing. In addition to this, the growth of temperature is associated not only with the increasing volume of greenhouse gases, but also with dissipation of the energy of catastrophic processes in the Earth’s crust and in the atmosphere. Naturally, during dissipation, these processes must make a significant contribution to the internal energy of the Earth’s crust, increasing its temperature. Unfortunately, practically no one studies these processes. Increase in temperature due to deformation and sea wave processes can occur not only in the Earth’s crust, but also in the atmosphere, as noted in [1]. The occurrence of microbaroms correlates with the occurrence of microseisms. Moreover, the same waves excite microseisms and microbaroms—these are gravity sea waves. The energy of microseisms depends on the energy of sea waves. In [2], the connection between microseisms and specific storms was studied. Large storms significantly increase microseismic energy [3,4,5], partial dissipation of which in the marine crust leads to an increase in the temperature of the World Ocean. In this regard, variation of seismic noise on decadal scales was studied for its relationship with climate variability [6].

In our assessments, we must take into account the role of deformation processes in the Earth’s crust, which cause increased emissions of natural gases from the mantle and crust into the atmosphere. In order to study the connection between the deformation processes of the Earth’s crust and emissions of natural greenhouse gases, a hard- and software controlled complex for gas-strain monitoring of the transition zone “atmosphere-hydrosphere-lithosphere” was created at Shultz Cape, the Primorsky Territory, Russia, and on the shelf of the Sea of Japan.

## 2. Hard- and Software Controlled Complex for Gas-Strain Monitoring

The hard- and software controlled complex consists of laser interference systems for measuring deformations of the Earth’s crust and variations in atmospheric pressure, stationary gas analyzing equipment and a weather station located at Shultz Cape, and also meteorological equipment and gas analyzing equipment located on the research vessel (R/V) “Professor Gagarinskiy”. When conducting the experiment to study the connection between the dynamics of greenhouse gases in the shelf area of the Sea of Japan and variations in the deformations of the Earth’s crust over a certain period of time, the R/V “Professor Gagarinskiy” was anchored on the shelf of the Sea of Japan, 1.8 km from the location of the laser strainmeter, see Figure 1.

### 2.1. Laser-Interference Complex

On the Gamov Peninsula, the Primorsky Territory, with the center in the point with coordinates 42.58 °N. and 131.15 °E. on the territory of the Marine Experimental Station (MES) “Shultz Cape”, a laser-interference measuring complex of instruments is located, including a laser nanobarograph, laser strainmeters with measuring arm lengths of 52.5 and 17.5 m, and a laser meter of hydrosphere pressure variations (see Figure 2).

The laser nanobarograph [7] was created to study the relationship between processes in the atmosphere, lithosphere, and hydrosphere. A block of aneroid boxes, used in classic pointer barometric instruments, is used as a measuring sensor of atmospheric pressure in the laser nanobarograph. In order to increase sensitivity when recording movement of the loose end of the aneroid boxes block, laser interference methods are used, based on an equal-arm Michelson interferometer with measuring (and “reference”) arm length of 20 cm.

The radiation source is a frequency-stabilized helium-neon laser by Melles Griot, which ensures frequency stability to the ninth decimal place. The laser nanobarograph scheme also includes a block of aneroid boxes with mirror coating, a digital recording system, and a block for transferring the obtained experimental data to the experimental database. The general view of the laser nanobarograph is shown in Figure 3. Its main technical characteristics are as follows: operating frequency range from 0 (conditionally) to 1000 Hz; accuracy of measuring variations in hydrosphere pressure—up to 1 mPa.

The dynamic range of all interferometers is significantly expanded by way of the level reset system and a feedback system that controls the interferometers operation. The 52.5 m and 17.5 m laser strainmeters form the basis of the two-dimensional laser strainmeter described in [8]. The optical elements of each laser strainmeter are mounted on two granite abutments fixed to the ground rocks. One abutment of the 52.5 m laser strainmeter is mounted on natural granite rock, and the other abutment stands on high-density loam. The height of the first abutment is about 1 m, and the height of the second is about 3 m. All abutments are cone-shaped, widening towards the bottom. Both abutments of the 17.5 m laser strainmeter, about 1.5 m high, are mounted on loam. Figure 4 shows a photograph of the interference unit of the 52.5 m laser strainmeter.

Figure 5 shows the interference unit of the 17.5 m laser strainmeter. In 2022, large-scale work was carried out to modify various components of strainmeters, since the devices have quite large spatial dimensions. In 2023, some work continued on additional thermal stabilization of the premises of laser interference installations. The main technical characteristics of laser strainmeters are as follows: operating frequency range from 0 (conditionally) to 1000 Hz; accuracy of measuring displacements of the Earth’s crust is 0.01 nm.

### 2.2. Closed-Type Stationary CO_2_/H_2_O Gas Analyzer LI-7200 RS

In 2023, at MES “Shultz Cape”, an Eddy Covariance station was installed based on a closed-type CO_2_/H_2_O gas analyzer LI-7200RS, model LI-7200RSF, LI-COR (Lincoln, NE, USA), air flow module 7200-102 (Figure 6a). The station is equipped with a SmartFlux 2 data processing system for Eddy Covariance systems in real time. The measuring frequency of the gas analyzer is from 5 to 20 Hz. The CO_2_ measurement range is from 0 to 3000 ppm. The sensitivity of CO_2_ measurements (mol H_2_O/mol CO_2_) is ±0.02. The station is equipped with a high-precision digital 3-axis (3D) ultrasonic anemometer Gill Windmaster 3D for measuring turbulent flows. The main characteristics of the anemometer are as follows: measuring speed (data output frequency)—20 Hz; range of measured wind speeds—0–45 m/s; resolution—0.01 m/s, error (RMS)—1.5% at 12 m/s; operating range of wind directions—0–359°; resolution—0.1°, error—2° at 12 m/s; body material—aluminum/carbon fiber.

The atmosphere is a variable environment, where continuous changes occur in the gas composition, aerosols, and meteorological elements (temperature, pressure, air humidity, wind direction, and speed), stimulating creation and operation of various observational systems for monitoring the atmosphere composition. Equipment by the manufacturer LI-COR—the closed-type CO_2_/H_2_O gas analyzer LI-7200RS—is specially designed for high-speed and high-precision measurements of CO_2_ and water vapor concentrations in the atmosphere (Figure 6b). It combines the advantages of closed-type gas analyzers (high accuracy of measurements, productivity, compact size, and resistance to the most adverse conditions) with the low power consumption of open-type gas analyzers.

The LI-7200RS gas analyzer takes advantage of non-dispersive infrared spectroscopy to quickly and reliably measure the density of CO_2_ and water vapor in the surrounding atmosphere (wavelength of 4260 nm by CO_2_ and 2590 nm by water vapor) [9]. A solid-state laser diode at the base of the sensor produces infrared radiation that passes through a system of thermally controlled optical filters and a confined air sample space and is applied to a thermally controlled lead selenide detector (Figure 6c). Some infrared rays are absorbed by CO_2_ and water vapor in the sample space. Gas concentrations are calculated by the ratio of absorbed IR radiation to the reference value.

Owing to the patented system, the gas analyzer also makes high-speed measurements of temperature and pressure in the sampled air in parallel with analysis of the gas composition. For each air sample, all of these data are combined and analyzed by EddyPro Software 7.0.9, producing wet and dry mole fraction values. The analyzer can make up to 20 measurements per second.

The obtained experimental data from the stationary gas analyzer after pre-processing, filtration, and decimation, are recorded in the created experimental database with a sampling period of 30 s.

The operation of all measuring devices is accompanied by mobile GPS receivers, which allows further synchronization of all received data in time.

### 2.3. Shipboard Gas Analyzing and Meteorological Equipment

The gas analyzing equipment consists of the following: a gas analyzer, Picarro G2311-f for high-precision continuous measurements of carbon dioxide, methane, and water vapor in the surface layer of the atmosphere; and a chromatographic gas complex “CRYSTALLUX 4000M” for discrete measurements of the content of methane, hydrocarbon gases, carbon dioxide, nitrogen, oxygen in the atmosphere, and sea water column.

Atmochemical measurements were carried out using a Picarro G2311-f laser analyzer (Picarro, Santa Clara, CA, USA), based on the WS-CRDS system (Wave Length Scanned Ring Down Spectroscopy)—light absorption spectroscopy in multi-stream non-axial cells when scanning along wavelengths) [10,11]. The measuring speed is 10 Hz. The measurement range for CO_2_ is 300–500 ppm, and for CH_4_, it is 1–3 ppm [11,12]. The analyzer is capable of automatically correcting the influence of water on measurements and assesses the influence of spectral interference. The instrument was calibrated annually during 2018–2022, and it was also calibrated immediately before the voyage, using certified gas standards for CO_2_ in the range of 360–500 ppm and for CH_4_, 1–3 ppm.

The gas analyzer, located in the upper deck laboratory, was equipped with a vacuum pump for continuous pumping of outside air, and with air intake devices of our own design (Figure 7). The air intake chamber of the analyzer was located on a projecting rod in the front part of the vessel at 10 m above sea level.

The meteorological equipment complex installed on the vessel consists of the following: Vaisala Weather Transmitter WXT520 (Helsinki, Finland) meteorological complex (temperature, humidity, pressure, wind speed and direction, precipitation level), Kipp&Zonen CNR4 Net Radiometer (total, short-wave, long-wave, reflected and effective radiation, albedo, and other parameters), photosynthetic radiation sensor LICOR LI190SB PAR Quantum Sensor (Lincoln, NE, USA) (measurement of photosynthetic radiation (PAR)), data logger CR1000, and Campbell Scientific (commutation of all meteorological equipment and recording in continuous mode with 1 min averaging and saving to a separate computer).

As a result of the measurements, an array of continuous data was obtained, with a recording frequency of 10 Hz, for 42 days of the expedition. All primary data were filtered, taking into account the influence of the ship’s exhaust gases. A joint analysis of CO_2_ concentrations, course of the ship, and wind direction (true wind direction) allows to effectively carry out such data screening. Next, the entire data array was averaged over 1 and 5 min time intervals and combined with meteorological parameters.

## 3. Development of Gas-Strain Monitoring Methodology at Shultz Cape

After installation and commissioning, the measuring complex, consisting of a gas analyzer, laser strainmeters, a laser nanobarograph and a weather station, the complex was put into operation to perform continuous measurements of variations in the Earth’s crust deformations, various meteorological parameters, and concentrations and flows of carbon dioxide and water vapor. After preliminary processing, the obtained information was entered into the created experimental database and was further processed using statistical and spectral estimation methods. To study possible connections between variations in deformations of the Earth’s crust and variations in the concentration of carbon dioxide and water vapor, we will compare the data obtained from laser interference systems and the gas analyzer with further study of general patterns in the behavior of the Earth’s crust deformations and variations in greenhouse gas concentrations.

Figure 8 shows graphs of variations in the Earth’s crust deformations, atmospheric pressure, and concentrations of carbon dioxide and water vapor.

In recent years, the content of greenhouse gases (GHGs) in the atmosphere has reached new record levels every year, and the upward trend continues. According to NOAA background monitoring data at the Mauna Loa Observatory (Hawaii, HI, USA), the global average CO_2_ concentration in January 2024 was already 422.8 ppm, and the average annual concentration in 2022 was 417.9 ppm [13]. According to Roshydromet, the average annual CO_2_ value at stations in the Russian Federation located under background conditions exceeded 415 ppm back in 2020, and in areas exposed to the influence of regional sources, the increase in CO_2_ concentration reached 419–428 ppm [14]. In general, the CO_2_ content in winter in the area of MES “Shultz Cape” varied from 424.3 to 449.1 ppm, with an average of 432.2 ppm. In general, this level of CO_2_ concentrations quite clearly illustrates the increased level of its content in the atmosphere during the winter period. This fact is due to both the peculiarities of atmospheric transport (mainly from the continental part) and the almost complete slowdown in the processes of photosynthesis and plant vegetation (sink of carbon dioxide is difficult).

As we can see from these graphs, the concentration of carbon dioxide (December 2023) in the surface layer of the atmosphere on average increases with time. Similar behavior is observed for graphs of variations in displacements of the Earth’s crust sections recorded by the laser strainmeters and in atmospheric pressure variations recorded by the laser nanobarograph. Only water vapor data behaves differently. Next, the obtained data were subjected to spectral processing using the periodogram method with the number of averages set to 3. The results of spectral processing of some of the data presented in Figure 8 are shown in Figure 9.

As we can see from all the presented spectra, the maxima corresponding to the diurnal and semidiurnal tides are distinguished from the rest. In the spectra of records of the laser strainmeters and the laser nanobarograph, a powerful peak with period of about 8 h stands out. We failed to identify higher-frequency components, since we have data with sampling period of 30 min, which allows us to obtain information about oscillations with periods of more than 1 h.

## 4. Development of Gas-Strain Monitoring Methodology on the Shelf of the Sea of Japan

Experimental studies on the shelf took place in different weather conditions, which led to some distortion of the obtained experimental data, therefore, a limited series of data with duration from 13 h 50 min on 22 November to 3 h 37 min on 29 November was selected for processing and analysis. On the ship, variations in the concentration of methane, carbon dioxide, and water vapor were measured; and at Shultz Cape, deformations of the Earth’s crust were measured using laser strainmeters, and variations in atmospheric pressure were measured using the laser nanobarograph. Figure 10 shows almost synchronous experimental data on variations in the concentration of methane, carbon dioxide, water vapor, microdeformations of the Earth’s crust, measured by the 17.5 m laser strainmeter, and variations in atmospheric pressure, measured at Shultz Cape by the laser nanobarograph.

Analysis of the dynamics of greenhouse gas concentrations in the surface layer of the atmosphere (November 2022) showed significant variability. CO_2_ concentrations during the multi-day station varied from 419.6 to 443.2 ppm, with an average of 429.6 ppm and a median of 425.9 ppm. CH_4_ concentrations ranged from 2.02 to 2.27, with an average of 2.08 ppm and a median of 2.05 ppm.

Upon visual inspection of these graphs, we can note that in the records of variations in carbon dioxide concentration there are many anomalous emissions that may be associated with the influence of life activities on the ship. Variations in water vapor and deformations of the upper layer of the Earth’s crust do not correlate well with variations in other parameters. Almost synchronous anomalous behavior is observed in atmospheric pressure variations and variations in methane concentration. An abrupt change in atmospheric pressure correlates with an abrupt release of methane. With a leap in atmospheric pressure of about 6700 Pa, the magnitude of the methane emission pulse was 0.19 ppm, which is almost four times greater than the background value. We can also note that these sudden emissions of methane and the jump in atmospheric pressure correlate with an abrupt decrease in humidity of about 25%.

Next, let us analyze the changes in the graphs shown in Figure 10 in the infragravity range, i.e., in the range of periods from 1 to 50 min. In the range of periods from 1 to 10 min in the beginning of observation (22 November), during spectral processing of laser strainmeters and the laser nanobarograph records, powerful spectral components are observed at periods of 7 min 59.1 s, which sometimes decrease to periods of 7 min 52.6 s and 7 min 45.5 s. As a typical example, Figure 11 shows the spectra obtained by processing synchronous records of the laser strainmeter with measuring arm length of 52.5 m, the laser nanobarograph, and the laser strainmeter with measuring arm length of 17.5 m. In the spectrum obtained by processing the record of the 52.5 m laser strainmeter, the first peak corresponds to the period of 7 min 59.1 s, although the maximum is the peak with period of 6 min 55.1 s. In the spectrum obtained by processing the laser nanobarograph record, the maximum peak with the period of 7 min 59.1 s stands out. In the spectrum obtained by processing the record of the 17.5 m laser strainmeter, the maximum peak is observed with the period of 9 min 28.9 s, and the second largest peak is associated with the period of 7 min 59.1 s. When comparing the obtained spectra shown in Figure 11, we can argue that all three instruments located at Shultz Cape recorded oscillations with period of about 7 min 59.1 s. Although, as we mentioned above, the period of these oscillations changes over time within certain limits, which may be associated both with natural processes, for example, with the phenomenon of non-isochronism, and with the effect of processing, when lower-frequency processes of large amplitude, and even trends, could influence the precise determination of the periods of these oscillations.

Towards the end of the experiment (28 November), powerful spectral components in this range of periods were observed not at Shultz Cape, but in the area where the R/V “Professor Gagarinskiy” was stationed. Figure 12 shows the spectra obtained by processing synchronous sections of records based on variations in the concentrations of methane, carbon dioxide, and water vapor, where powerful spectral components are identified at the periods of 7 min 45.5 s, 7 min 38.5 s and 7 min 38.5 s, respectively. During the same observation period, in this range of periods, peaks at the period of 8 min 40.7 s are identified from the records of laser strainmeters and the laser nanobarograph. This transformation of oscillation periods is unclear. It turns out that oscillations with higher-frequency periods were displaced by lower-frequency processes from the territory of Shultz Cape within several days. As we can see from graphs shown in Figure 12a–c, a powerful peak with periods of about 7 min 45.5 s and 7 min 38.5 s stands out everywhere. Considering that when processing a series with duration of 512 points and at sampling frequency of 0.016(6) Hz in the spectrum obtained on the basis of the fast Fourier transform, harmonics with periods of 7 min 45.5 s and 7 min 38.5 s are close, we can state that this is the same peak related to the same natural process.

Considering this circumstance, and also the graphs shown in Figure 11, we can assume that these fluctuations in variations in the upper layer of the Earth’s crust deformations at Shultz Cape, fluctuations in atmospheric pressure at Shultz Cape, and fluctuations in the concentration of methane, carbon dioxide, and water vapor at sea, several kilometers from Shultz Cape, are caused by the same processes. This process can most likely be attributed to atmospheric oscillations in the infragravity range. We can, however, note that these oscillations are not always observed simultaneously at installations located in the same place. For example, Figure 13 shows the spectra obtained by processing synchronous records of methane, carbon dioxide, and water vapor concentrations. When comparing the graphs shown in Figure 13a,b, a powerful peak is observed with period of 7 min 45.5 s, but there is no such peak on the graph of water vapor concentration, but the peak is observed at period of 6 min 44.2 s. That is, infragravity fluctuations do not always cause synchronous corresponding fluctuations in the measured parameters of the geospheres.

In the range of periods from 10 to 20 min in the records of the 52.5 m laser strainmeter and laser nanobarograph, and the records of carbon dioxide concentrations, oscillations can be distinguished at periods of 18 min 17.1 s. The 17 min 39.1 s fluctuations are present mainly in the records of methane concentrations, and sometimes in the records of carbon dioxide concentrations, water vapor, and atmospheric pressure variations at Shultz Cape. Oscillations of 17 min 04.0 s can be traced in the laser nanobarograph and water vapor records. In the range of periods from 20 to 30 min, powerful synchronous oscillations stand out at the maximum, corresponding to the period of 20 min 28.8 s. These peaks stand out when processing records from both laser strainmeters and the laser nanobarograph; large peaks with this period stand out when processing records of water vapor concentration. We can also note neighboring maxima with periods of 21 min 19.1 s, which are observed at separate time intervals in the spectra isolated when processing records from the 52.5 m laser strainmeter, laser nanobarograph, and records of methane and carbon dioxide concentrations. Oscillations with period of 25 min 35.1 s are isolated from the records of laser strainmeters, of laser nanobarograph and water vapor records. Oscillations with the period of 28 min 26.7 s are isolated from the records of laser strainmeters, laser nanobarograph, concentrations of methane, carbon dioxide and water vapor records. In the lower frequency range, when processing records from all instruments, a maximum with a “wandering” period from 30 min 07.6 s to 31 min 59.1 s is identified, which is found both in the spectra of methane, carbon dioxide, and water vapor concentrations, and in the spectra of the laser strainmeters and laser nanobarograph records. At the same time, in the spectra of methane, carbon dioxide, and water vapor concentrations, their intensity is noticeably greater, see, for example, Figure 14. For the spectra obtained by processing data on concentrations of methane and water vapor, the peak with the period of 31 min 59.1 s has the maximum value, and in the spectrum of carbon dioxide it is the second largest.

## 5. Conclusions

At Shultz Cape, the Primorsky Territory, Russia, a stationary gas-strain monitoring system was launched, consisting of a stationary gas analyzer, laser strainmeters, a laser nanobarograph and a weather station, designed to study the correlations between the deformation of the Earth’s crust and the concentration and flow of climate-active gases. When processing and analyzing the first obtained data, it was determined that an increase in deformation of the upper layer of the Earth’s crust correlates with an increase in carbon dioxide concentrations and an increase in atmospheric pressure. In addition, the spectra of records of the Earth’s crust deformations, variations in atmospheric pressure, and carbon dioxide content contain powerful spectral components corresponding to the diurnal and semidiurnal tides.

When analyzing data obtained from variations in the Earth’s crust deformations and atmospheric pressure at Shultz Cape, and from variations in the concentrations of methane, carbon dioxide, and water vapor on the shelf of the Sea of Japan, the following patterns were established: An abrupt change in atmospheric pressure correlates with an abrupt release of methane. With a leap in atmospheric pressure of about 6700 Pa, the magnitude of the methane emission pulse was 0.19 ppm, which is almost four times greater than the background value. We can also note that these sudden outbursts of methane and a leap in atmospheric pressure correlate with an abrupt decrease in humidity by about 25%.

-In the spectra of the laser strainmeters and laser nanobarograph records, powerful “wandering” peaks with periods ranging from 7 min 59.1 s to 7 min 45.5 s are observed, which after some time stand out in the records of variations in the concentrations of methane, carbon dioxide, and water vapor; at the same time, in the spectra of deformations of the Earth’s crust and atmospheric pressure, peaks with slightly longer periods stand out at Shultz Cape. We can assume that these fluctuations are caused by some atmospheric depression, which over the course of several days slowly shifted from Shultz Cape to the location of the R/V “Professor Gagarinskiy”, where atmospheric spatial irregularities increased in size.-The origin of oscillations in the lower frequency range (10–30 min) is difficult to explain without additional experiments. Nevertheless, we can assume that the maximum, with a period of 20 min 28.8 s, can be conditioned by the spheroidal tone 0S0, which, in accordance with [6], causes fluctuations in atmospheric pressure of a similar length.-Fluctuations of long periods (from 30 min 07.6 s to 31 min 59.1 s) were previously identified in the records of sea level fluctuations during an atmospheric pulse’s passage over the Sea of Japan, resulting from the explosion of the Hunga Tonga-Hunga_Ha’apai volcano [15]. The origin of this oscillation was first explained by the excitation of one of the seiches of the Sea of Japan, but later it was found that these oscillations are caused by eigen oscillations of the ionosphere, i.e., Lamb waves. Changes in the periods of these oscillations are associated with changes in the size of the corresponding layers of the troposphere.

Preliminary methodological measurements of greenhouse gas variations on the shelf of the Sea of Japan and Shultz Cape, synchronously with variations in deformation of the Earth’s crust, showed the high performance of the created hard- and software controlled gas-strain complex, aimed at studying the nature of greenhouse gases associated with deformation processes in the Earth’s crust and mantle.

## Figures and Tables

**Figure 1 sensors-24-02602-f001:**
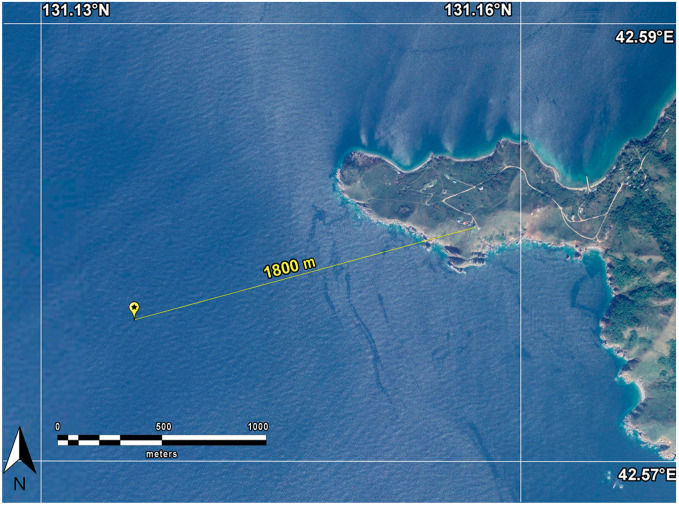
Map showing R/V “Professor Gagarinsky” anchorage location.

**Figure 2 sensors-24-02602-f002:**
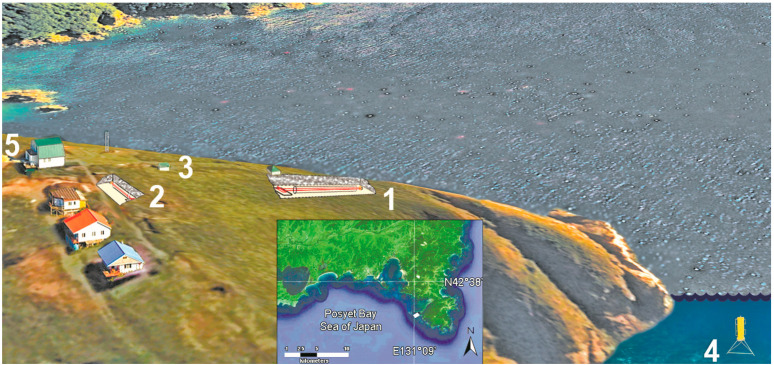
Arrangement of laser strainmeters: 1—with measuring arm of 52.5 m; 2—with measuring arm of 17.5 m; 3—laser nanobarograph; 4—laser meter of hydrosphere pressure variations; and 5—laboratory building.

**Figure 3 sensors-24-02602-f003:**
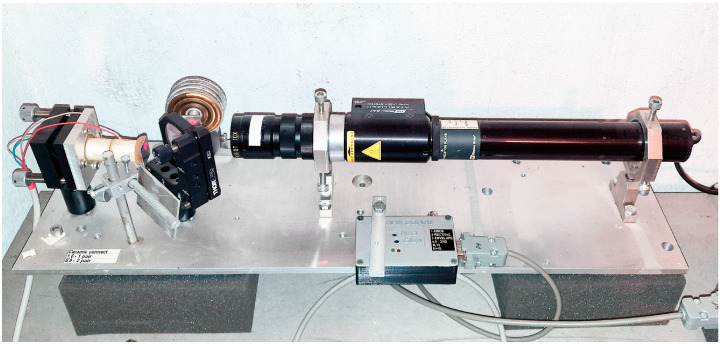
General view of the laser nanobarograph.

**Figure 4 sensors-24-02602-f004:**
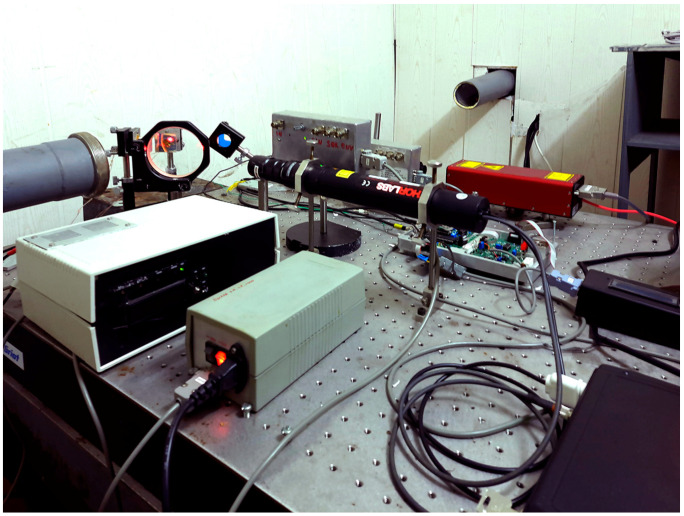
Central interference unit of the 52.5 m laser strainmeter.

**Figure 5 sensors-24-02602-f005:**
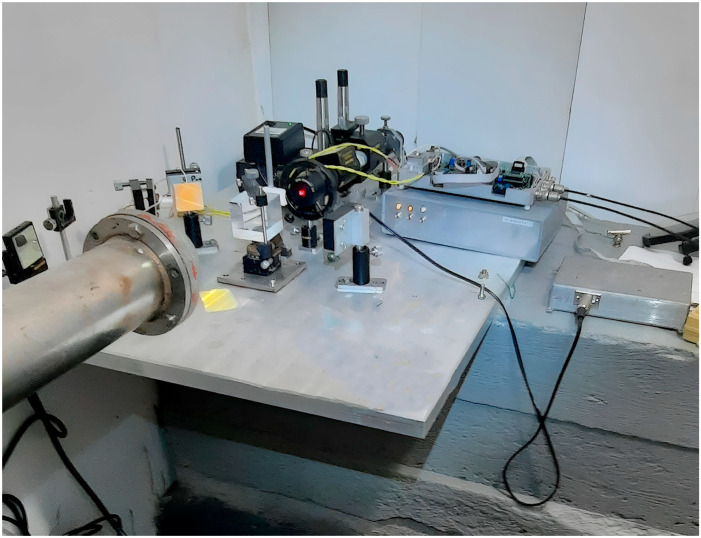
Central interference unit of the 17.5 m laser strainmeter.

**Figure 6 sensors-24-02602-f006:**
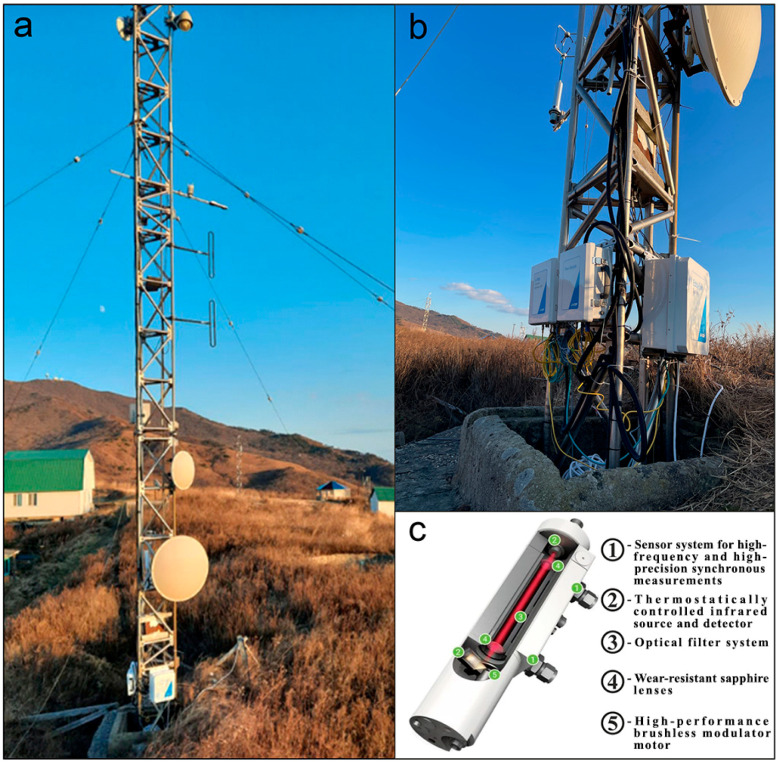
Eddy Covariance station in complex with weather sensors at MES “Shultz Cape” (**a**); closed-type CO_2_/H_2_O gas analyzer LI-7200RS (**b**); and gas analyzer structure diagram (**c**).

**Figure 7 sensors-24-02602-f007:**
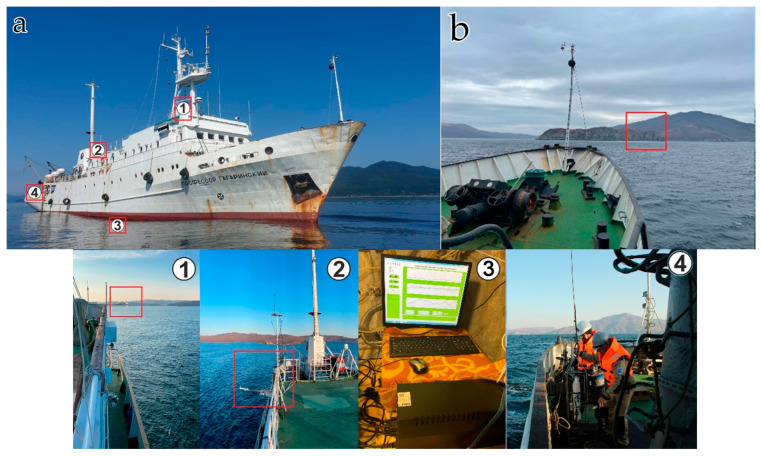
Research scheme. (**a**) shows a mobile shipboard measuring system based on the R/V “Professor Gagarinsky”: 1—atmochemical measuring complex (Picarro gas analyzer in the shipboard laboratory and position of the air intake device in the front part of the ship); 2—a set of meteorological instruments on the upper deck of the bridge; 3—shipboard gas analysis laboratory and ship flow system; and 4—deck outboard works. (**b**) shows hard- and software controlled complex for gas-strain monitoring at Shultz Cape.

**Figure 8 sensors-24-02602-f008:**
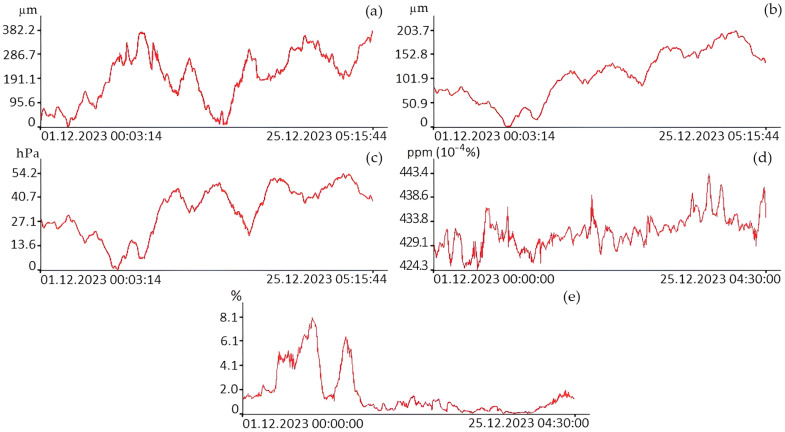
Time changes in the displacements of the measuring arm of the 52.5 m laser strainmeter (**a**), 17.5 m laser strainmeter (**b**), atmospheric pressure (**c**), carbon dioxide concentrations (**d**), and water vapor (**e**).

**Figure 9 sensors-24-02602-f009:**
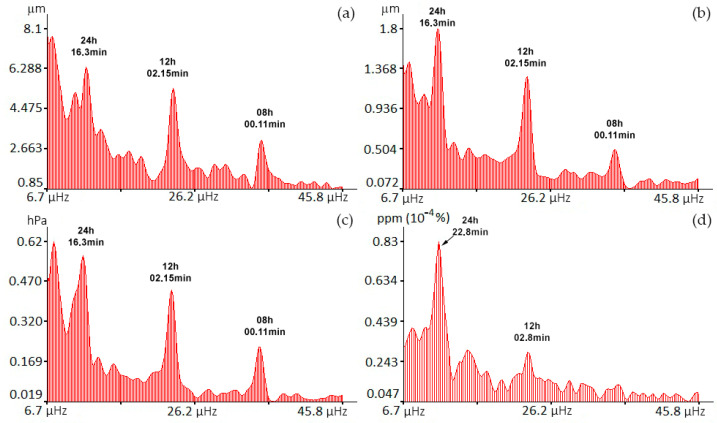
Spectra obtained by processing data from 52.5 m laser strainmeter (**a**), 17.5 m laser strainmeter (**b**), laser nanobarograph (**c**), and gas analyzer (**d**)—carbon dioxide.

**Figure 10 sensors-24-02602-f010:**
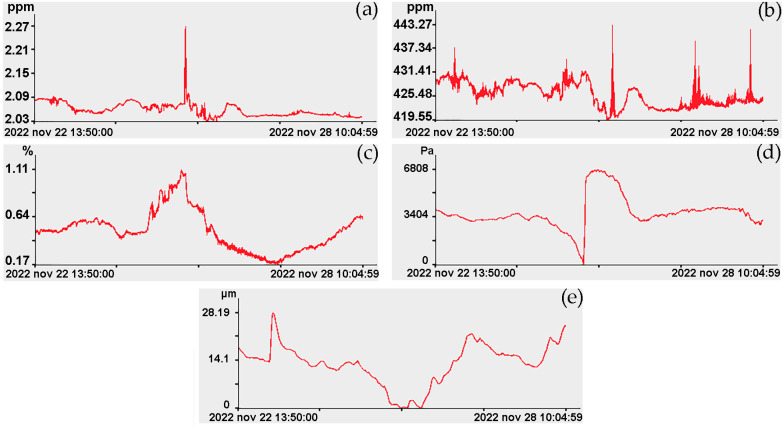
Synchronous record sections: (**a**) methane concentrations; (**b**) carbon dioxide concentrations; (**c**) water vapor concentrations; (**d**) variations in atmospheric pressure at Shultz Cape; and (**e**) deformations of the upper layer of the Earth’s crust at Shultz Cape.

**Figure 11 sensors-24-02602-f011:**
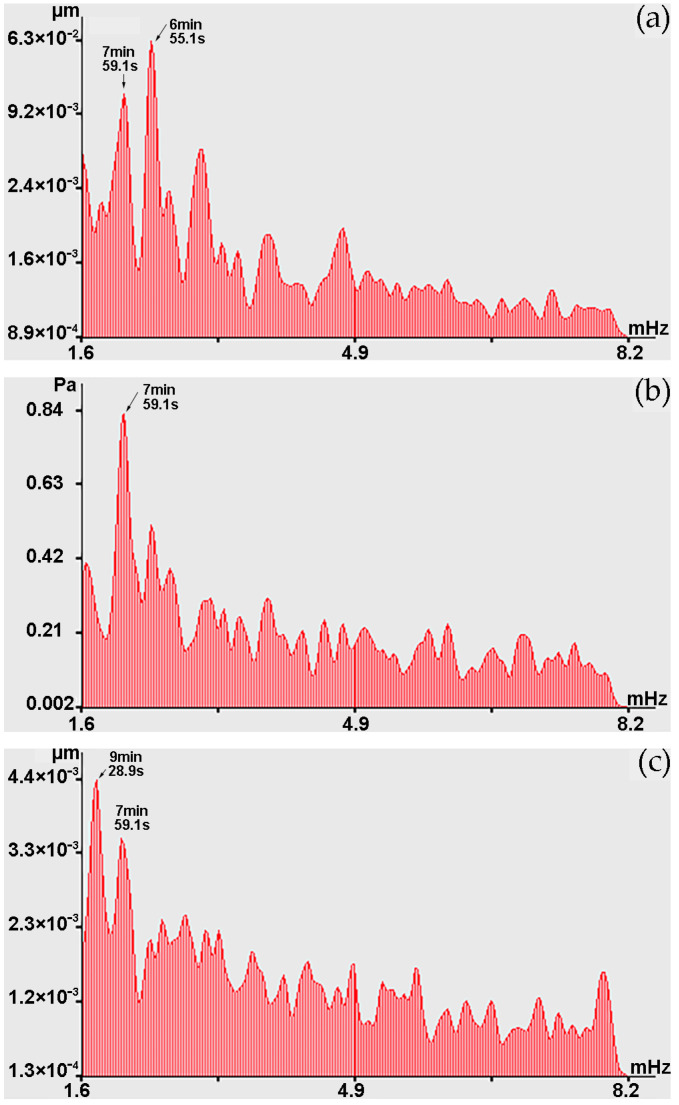
Spectra obtained by processing synchronous sections of records: (**a**) 52.5 m laser strainmeter; (**b**) laser nanobarograph; and (**c**) 17.5 m laser strainmeter.

**Figure 12 sensors-24-02602-f012:**
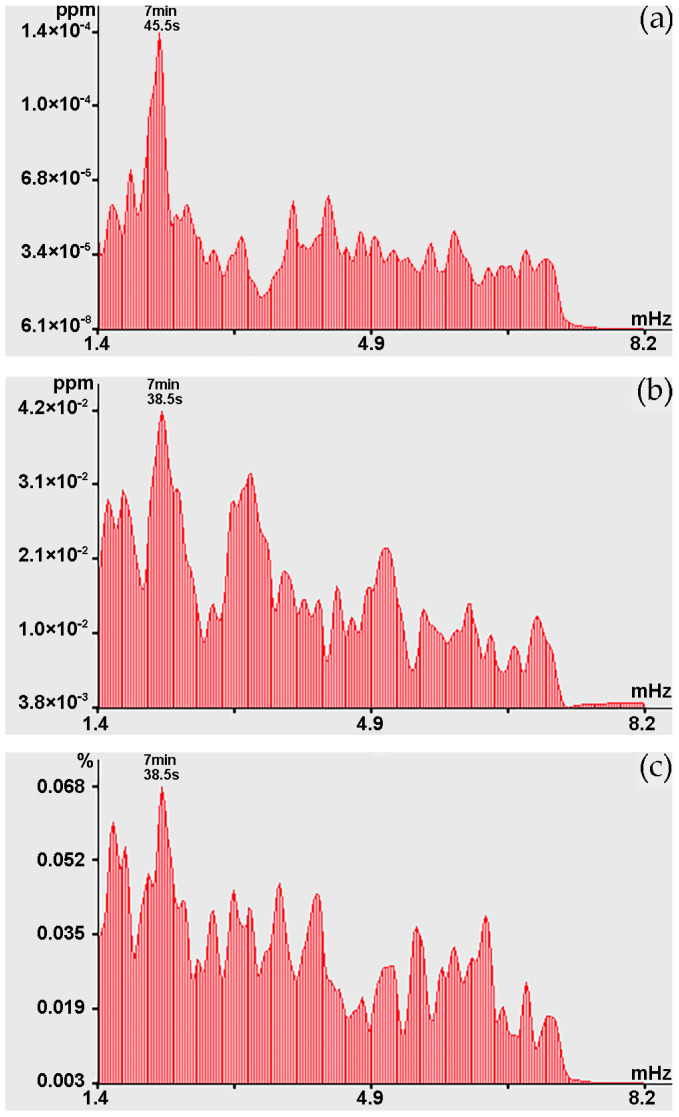
Spectra obtained by processing the record of variations in concentrations: (**a**) methane; (**b**) carbon dioxide; and (**c**) water vapor.

**Figure 13 sensors-24-02602-f013:**
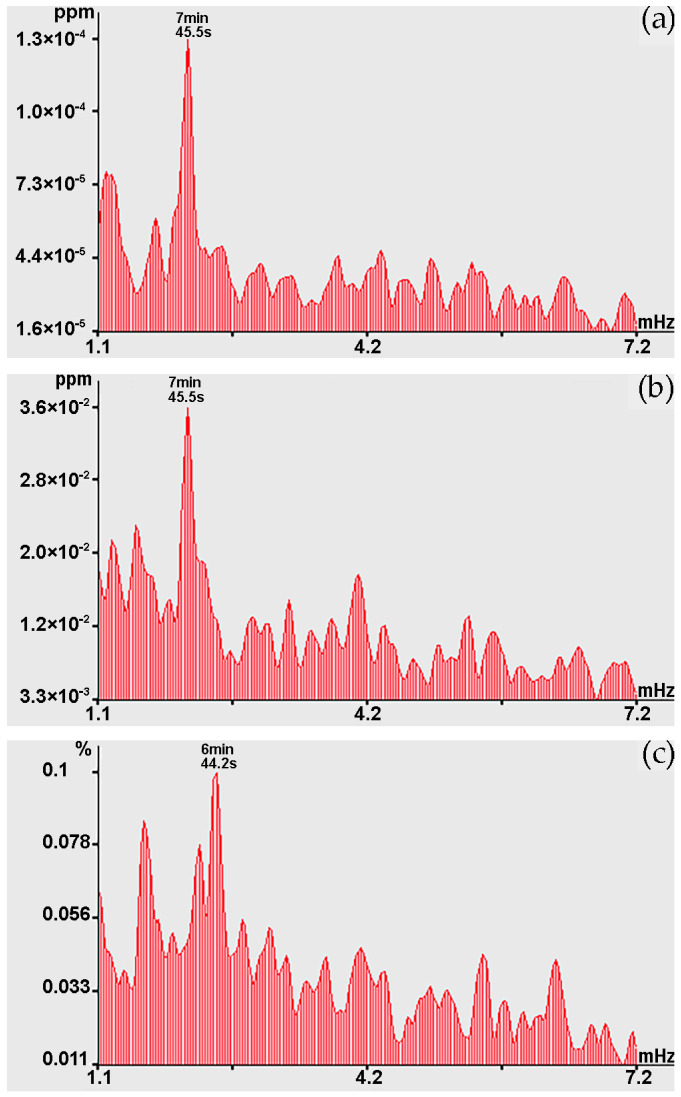
Spectra obtained by processing the record of concentration variations: (**a**) methane; (**b**) carbon dioxide; and (**c**) water vapor.

**Figure 14 sensors-24-02602-f014:**
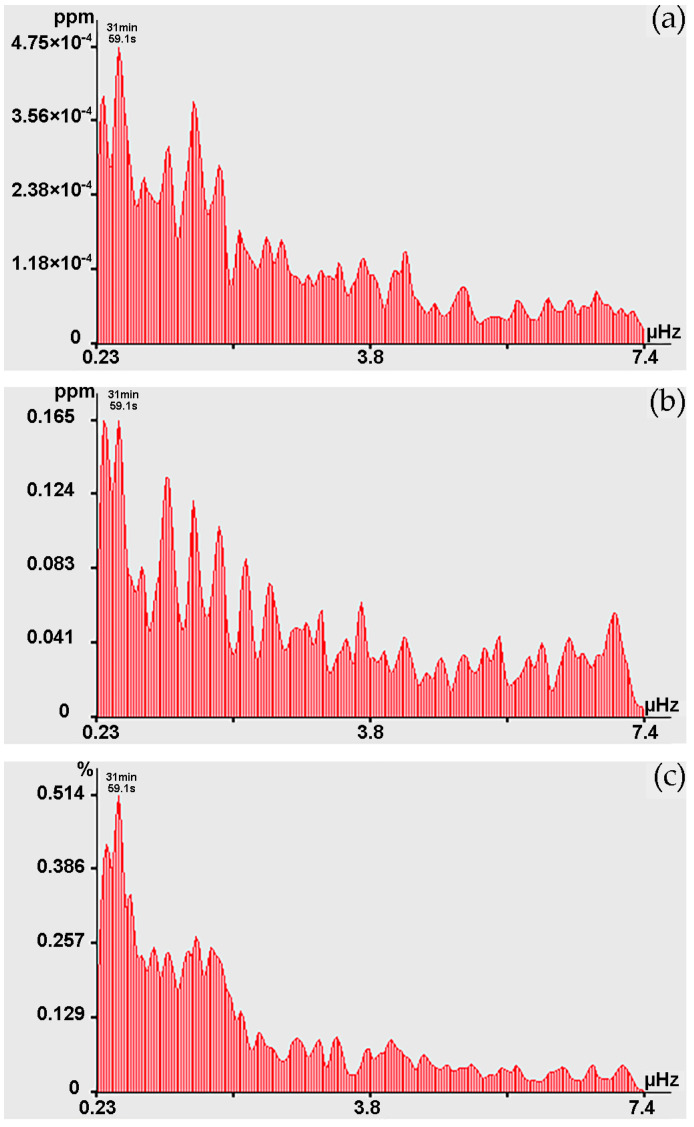
Spectra obtained by processing the record of variations in concentrations (0.23–7.4 µHz): (**a**) methane; (**b**) carbon dioxide; and (**c**) water vapor.

## Data Availability

Third Party Data. Restrictions apply to the availability of these data. The laser strainmeter presented in the study is included in the Appendix A.

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
