# Peer review of "Hard- and Software Controlled Complex for Gas-Strain Monitoring of Transition Zones"

_sensors, 2024, doi:10.3390/s24082602_

Round 1
Reviewer 1 Report
Comments and Suggestions for Authors
The paper employs a complex, comprehensive integrated measurement system of instruments to study the correlations between deformation of the Earth’s crust and concentration and flow of climate-active gases. The paper reports the research results and conclusions, affirming that the research content is worthy of publication in this journal.
Only two minor issues are provided for the authors to consider:
1. In the abstract, it is suggested to be more specific and clear in the terminology related to the high-frequency range.
2. In line 124, the ninth digit, it is recommended to be more specific,in hertz.
3. In line 179, what is the wavelength of the laser?
Author Response
Thanks for the comments for the development of the study.
Only two minor issues are provided for the authors to consider:
- In the abstract, it is suggested to be more specific and clear in the terminology related to the high-frequency range.
Response 1: Corrected. The definition was changed.
- In line 124, the ninth digit, it is recommended to be more specific,in hertz.
Response 2: The terminology has been corrected
- In line 179, what is the wavelength of the laser?
Response 3: Added laser characteristics. Wavelength of 4260 nm by CO2 and 2590 nm by water vapour.

Reviewer 2 Report
Comments and Suggestions for Authors
In this article, the authors summarise and discuss a hardware and software control complex for gas strain monitoring in the transition zone. The high performance of the created hardware and software controlled gas strain complex is aimed at studying the nature of greenhouse gases associated with formation processes in the crust and mantle. It provides new insights and ideas for subsequent innovations in related directions. Therefore, this manuscript can be considered for publication only after minor revisions.
The following comments will be helpful for authors to improve the manuscript further:
1) The reference and figure note formatting needs to be revised again, for example, the chemical molecules in references 11, 12 and figure 6 are written in an incorrect format.
2) There are flaws in the writing of the content of the article, for example, the writing of the time in lines 359 and 372; also try to be uniform in the expression of time throughout the article, try not to have a time that has both minutes and seconds.
3) The English expressions and picture formats used throughout the manuscript should be significantly improved.
4) More detailed explanations should be provided on the effects of interfering gases, temperature, and moderation on materials.
5) In the record of CO2 concentration changes, more scientific explanations or analytical charts should be provided for explanation.
Comments on the Quality of English LanguageMinor editing of English language required
Author Response
Thanks for the comments for the development of the study.
1) The reference and figure note formatting needs to be revised again, for example, the chemical molecules in references 11, 12 and figure 6 are written in an incorrect format.
Response 1: The format of chemical molecules has been corrected
2) There are flaws in the writing of the content of the article, for example, the writing of the time in lines 359 and 372; also try to be uniform in the expression of time throughout the article, try not to have a time that has both minutes and seconds.
Response 2: To identify natural oscillations that have periods greater than 60 seconds, it is considered most presentable to specify the period of oscillation in minutes and seconds.
3) The English expressions and picture formats used throughout the manuscript should be significantly improved.
Response 3: Some figures have been reworked. Corrected terminology.
4) More detailed explanations should be provided on the effects of interfering gases, temperature, and moderation on materials.
Response 4: The first conclusions about the correlation of some processes were made. The work is still in progress. It is now possible to investigate the interaction of processes with higher sampling rates.
5) In the record of CO2 concentration changes, more scientific explanations or analytical charts should be provided for explanation.
Response 5: More detailed explanations have been provided for comments 4 and 5.
